# Balanced Force Field ff03CMAP Improving the Dynamics Conformation Sampling of Phosphorylation Site

**DOI:** 10.3390/ijms231911285

**Published:** 2022-09-25

**Authors:** Bozitao Zhong, Ge Song, Hai-Feng Chen

**Affiliations:** 1State Key Laboratory of Microbial Metabolism, Joint International Research Laboratory of Metabolic and Developmental Sciences, Department of Bioinformatics and Biostatistics, National Experimental Teaching Center for Life Sciences and Biotechnology, School of Life Sciences and Biotechnology, Shanghai Jiao Tong University, Shanghai 200240, China; 2Shanghai Center for Bioinformation Technology, Shanghai 200240, China

**Keywords:** phosphorylation, force field, *ff03CMAP*, disordered proteins

## Abstract

Phosphorylation plays a key role in plant biology, such as the accumulation of plant cells to form the observed proteome. Statistical analysis found that many phosphorylation sites are located in disordered regions. However, current force fields are mainly trained for structural proteins, which might not have the capacity to perfectly capture the dynamic conformation of the phosphorylated proteins. Therefore, we evaluated the performance of *ff03CMAP*, a balanced force field between structural and disordered proteins, for the sampling of the phosphorylated proteins. The test results of 11 different phosphorylated systems, including dipeptides, disordered proteins, folded proteins, and their complex, indicate that the *ff03CMAP* force field can better sample the conformations of phosphorylation sites for disordered proteins and disordered regions than *ff03*. For the solvent model, the results strongly suggest that the *ff03CMAP* force field with the *TIP4PD* water model is the best combination for the conformer sampling. Additional tests of *CHARMM36m* and *FB18* force fields on two phosphorylated systems suggest that the overall performance of *ff03CMAP* is similar to that of *FB18* and better than that of *CHARMM36m*. These results can help other researchers to choose suitable force field and solvent models to investigate the dynamic properties of phosphorylation proteins.

## 1. Introduction

Post-translational modification (PTM) is common in nature [1]. It refers to the process by which a chemical group or small protein is covalently bound to an amino acid or protein. PTM plays a key role in mediating cellular signaling [2], regulation [3], and metabolic activities [4]. Therefore, the study of the mechanistic role of post-translational modifications has become a popular research field. Until now, more than 5000 types of PTM have been identified. Among all these types of PTM, the most common and pervasive one is phosphorylation [1]. It is estimated that one-third of all proteins in eukaryotic cells, including human cells, undergo phosphorylation and dephosphorylation mediated by roughly 500 putative kinases and 150 phosphatases. Phosphorylation modifications occur mainly on serine, threonine, and tyrosine, while some other residues can sometimes undergo O- and N-phosphorylation.

Reversible phosphorylation plays an important function in the cell. The reversible phosphorylation process is involved in signal transduction pathways, amplification, integration during the cell cycle, and reactions to external effectors. Moreover, specific phosphorylation controls protein turnover [5], localization (e.g., nuclear transportation [6]), chromatin remodeling [7], and the assembly of protein complexes [8,9]. In these biological processes, phosphorylation often functions by affecting residue–residue interactions. Compared with the 20 types of canonical residues, phosphorylated residues have novel physicochemical properties and negative charges to form electrostatic interactions. For example, the salt bridge between phosphoserine and lysine can stabilize the helical structure, especially when it is on the N-terminal end of the helix [10].

Previous statistical research found that many phosphorylation sites are located in disordered regions, which are preferred substrates of protein kinases [1]. Intrinsically disordered proteins (or intrinsically disordered regions) refer to peptides (or fragments) that lack a well-defined three-dimensional structure in solution under physiological conditions [11]. Phosphorylation in disordered proteins and regions is involved in the transition between disordered and folded states, changes in the association state, and the activation or deactivation of a protein [12,13,14,15]. For example, in the Ets-1 domain, phosphorylation can trigger disordered-to-folded transitions [16]. Recent research found that phosphorylation can also induce folding of the disordered translational inhibitor 4E-BP2 [14,17].

To detect phosphorylated protein structures and their regulation mechanisms, there are a series of experimental and computational methods that can be used. X-ray, NMR (Nuclear Magnetic Resonance), SAXS (Nuclear Magnetic Resonance) and FRET (Förster resonance energy transfer) can be used to detect phosphorylated protein structures [18]. The experimental measurements include atom coordinates, chemical shifts, scalar couplings, and so on. However, these experimental methods only reflect the average structural insights and have difficulty in tracing the motion of proteins, thus have the limitation of revealing mechanisms at the atomistic level.

Phosphorylation also plays a key role in plant biology [19]. In plant cells, many proteins undergo post-translational modifications and form the mature proteoforms that accumulate to form the observed proteome [20]. Plant PTM Viewer, a PTM database for plants, contains 437,404 PTMs matching 103,992 proteins from plants, of which 74.7% of modifications are phosphorylations [21]. In Arabidopsis, at least 7603 nonredundant proteins have been experimentally identified as phosphorylated at 42,649 different sites [19]. Phosphorylation can function as a molecular switch in regulation, and many signaling pathways work through phosphorylation. A well-studied example is the MAP kinase network, consisting of a hierarchy in which MAP triple-kinases phosphorylate and activate MAPK kinases, which in turn phosphorylate and activate MAPKs at conserved Thr and Tyr residues [22].

Using molecular dynamics (MD) to study the dynamical properties is an important complement to experimental methods [12]. Recently, many studies have used molecular dynamics simulations to study structural change via phosphorylation. Parameters for major types of phosphorylation (phosphorylated serine, threonine, and tyrosine) can be found in most well-known force fields. In the AMBER force field series, the ff9x family contains phosphorylated parameters from Homeyer et al. [23]. The AMBER *ff03* family has the post-translational modification parameter *Forcefield_PTM* from Duan et al. [24]. Another popular force field family, CHARMM, includes phosphorylated parameters for version 27, 36 and 36 m [25,26]. The Gromos force field family also includes two sets of modified residue parameters [27,28,29]. These force fields have already been used in modified protein structural research.

The accuracy of chosen force-field and phosphorylation parameters is the key for better dynamic simulation results. Unfortunately, recent studies carried out on phosphorylated residue parameters showed that the simulation result did not fit well with experimental data for dipeptides or disordered proteins. Vymetal et al. simulated the dipeptide structures of the top three phosphorylated amino acids with three commonly used force fields with their corresponding parameters for phosphorylation, the *ff99SBildn* and *ff03* in AMBER, and *CHARMM36m*. Results showed that dihedral angle distributions and the trends of the secondary structure changes for modifications were inconsistent [30]. Both the ratios in the α-helical, β-sheet and disordered regions, and the distribution of dihedral angles φ/ψ, showed significant inconsistency among the three force fields. For J-coupling constants, the results showed that the *CHARMM36m* force field had a greater difference from the experimental value compared to the other two force fields. Furthermore, all three force fields were not accurate enough. In another study, simulation of the SN15 peptides was carried out with *ff99SBildn* and *CHARMM36m*. The results of *CHARMM36m* were worse than those of *ff99SBildn* for SAXS values. However, the results of the AMBER force field were not perfect where the proportion of α-helix is higher than the actual conformation [31].

Considering that most of the popular major force fields (*ff99SBildn* [32], *ff03* [33] and *ff14SB* [34]) are more suitable for folded proteins, while phosphorylation sites are more likely to be located in disordered regions, we conjecture that the balanced force field *ff03CMAP* [35] might outperform other force fields in the simulation of phosphorylated proteins. Compared with the *ff03* force field, *ff03CMAP* adds a correction map (CMAP) parameter to adjust the dihedral angle potential energy, which affects the conformation of the proteins in simulation, achieving improvements in the simulation of disordered proteins without affecting the accuracy of folded proteins [35]. The CMAP method has been widely used to improve the accuracy of force fields [36,37], especially for disordered proteins [38,39,40,41,42,43,44]. The solvent model also affects protein conformations in simulations, therefore we also tested two water models of *TIP4P-Ew* [45] and *TIP4P-D* [46] with the two force fields to show a combination effect on phosphorylated protein simulation. To fully cover all kinds of phosphorylated proteins, we chose test systems ranging from dipeptides, disordered proteins, folded proteins, and disordered-folded protein complexes. This diverse test set can adequately cover most types of phosphorylated proteins. The results indicate that the combination between *ff03CMAP* and *TIP4PD* can better sample the conformations of phosphorylation sites for disordered proteins or disordered regions than *ff03*.

## 2. Material and Methods

### 2.1. Phosphorylated Protein Structural Propensity

At first, statistical analysis was carried out for all phosphorylated proteins in the PDB database [47]. This analysis was performed to determine the propensity of modification sites, which can help us to better understand the characteristics of the PTM protein structure.

In this statistical analysis, all proteins with phosphorylated serine, threonine, and tyrosine were collected. For each type of phosphorylation, the amount of phosphorylated protein and phosphorylated sites were calculated. We also compared the statistics with former research on the Swiss-Prot database [1]. We used the DSSP program [48,49] to assign the secondary structure of proteins and phosphorylation sites. In the DSSP prediction results, H, G and I were classified as helixes, B, E, T, S were considered as sheets, and blank was considered a disordered region. We calculated the protein secondary structure propensity by calculating the ratio between disordered residues and all residues. Differences between the secondary structure composition in protein and modification sites can present the structural tendency of modification sites.

### 2.2. Molecular Dynamics Simulation

After understanding the secondary structure tendency of post-translational modification sites, molecular dynamics simulations were conducted on various selected phosphorylated modified proteins, including dipeptides, disordered proteins, folded proteins and complexes of disordered and folded proteins. The test set covered all typical types of phosphorylated proteins. In simulation tests, we used 2 different force fields with 2 different solvent models to compare their performances on phosphorylated proteins. For force fields, we chose the widely-used *ff03* force field and the *ff03CMAP* force field, which has a well-balanced performance between folded and disordered proteins. For solvent models, *TIP4P-Ew* and *TIP4P-D* were chosen. In previous *ff03CMAP* simulation tests, *TIP4P-Ew* performed well in folded proteins while *TIP4P-D* performed better in disordered proteins. With the phosphorylated residue parameters from *Forcefield_PTM* [24], we can compare these 4 different methods in simulating modified peptides/proteins to determine the best force field and solvent model set in phosphorylated protein simulations. In addition, to obtain a broader view of the performance of *ff03* and *ff03CMAP*, we included the tests for two other force fields combined with their recommended water models: *CHARMM36m* [50] (with *TIP3P*) and *FB18* [51] (with *TIP3PFB*). The simulation systems and conditions are gathered in Table 1.

Initial structures of the G-X-G systems were built in the LEaP module in the AMBER 18 suite [61] and other structures were taken from the PDB database. All simulation systems were neutralized and solvated in octahedron boxes of *TIP4P-Ew* or *TIP4P-D* solvent models with 10 Å to the edge. We checked that the water boxes were big enough to keep the minimum distance between periodic images greater than the cutoff value, as shown in Appendix A. All bonds involving hydrogen atoms were constrained with the SHAKE algorithm [62]. The van der Waals interactions and short-range electrostatic interactions were truncated with 8 Å cutoff. The Particle-Mesh Ewald (PME) algorithm was used to calculate long-range electrostatic interactions [63]. Initial structures were relaxed by 3000 steps of minimization with the steepest descent method, followed by 3000 steps of minimization with the conjugate gradient algorithm. The minimized structures were then heated up to 298.15 K over 50 ps at constant volume, followed by another heating with the same temperature over 100 ps at a constant pressure of 1 bar, and equilibrated over 100 ps at the constant pressure. The production simulations were performed in the pmemd.cuda [64] with simulation time corresponding to Table 1. The simulation temperature was set to room temperature. As postulated in previous research [65,66], ionic concentration had significant impacts on the stability of solute’s intramolecular salt bridges and in turn changed the conformation ensemble. In this study, sodium ions were set to neutralized protein charges.

### 2.3. Evaluation Metrics

To compare the performance of various force fields and solvent models, we used a series of quantification metrics to evaluate simulation results. For dynamical properties, RMSD, RMSF and radius of gyration (Rg) were used for evaluation, and all of these metrics were calculated by CPPTRAJ in AmberTools [67]. For protein structural properties, chemical shifts, secondary structures and contact maps were used in comparison. Chemical shifts were calculated by SPARTA+ [68] for Cα, Cβ, C, N, Hα and HN atom types [68]. The detailed experimental data are listed in Appendix A. Secondary structure and contact maps were also calculated by CPPTRAJ in AmberTools [67]. The PyMOL molecular visualization system was used to present structures for all simulated proteins [69].

In addition, the quantification metric force field was used for the benchmark for all methods. The force field score was calculated as in Equation (1) [70]
(1)$${\mathrm{FF}}_{\mathrm{score}\;}=\frac{1}{N}\overset{i=1}{\underset{N}{\sum }}\frac{{\mathrm{FF}}_{\mathrm{rmsd}}}{{\mathrm{rmsd}}_{\mathrm{norm}\;}}$$
where N is the number of classes of experimental measurements, ${\mathrm{FF}}_{\mathrm{rmsd}}$ is the RMSD of the i  th class for simulated and experimental values, and ${\mathrm{rmsd}}_{\mathrm{norm}}$ is the lowest RMSD of i  th class in all force fields. Based on the force field score, which considers the overall properties of a protein, we crafted another metric called the modification site force field score to evaluate the experimental values corresponding to post-translational modification site. The modification site force field score was defined as in Equation (2)
(2)$${\mathrm{ModificationFF}}_{\mathrm{score}\;}=\frac{1}{N}\overset{i=1}{\underset{N}{\sum }}\frac{{\mathrm{ModificationFF}}_{\mathrm{rmsd}}}{{\mathrm{rmsd}}_{\mathrm{norm}\;}}$$
where ${\mathrm{ModificationFF}}_{\mathrm{rmsd}}$ is the RMSD of the i  th class for simulated and experimental values in modified residues, and ${\mathrm{rmsd}}_{\mathrm{norm}}$ is the lowest RMSD of i  th class among all force fields in modified residues. According to this metric, FFscore and ModificationFFscore are always larger than or equal to 1, and 1 is the best score theoretically.

## 3. Results

### 3.1. Structural Statistics of Phosphorylated Proteins in PDB Database

As we compared the secondary structure propensity of phosphorylation modifications in PDB, we found distinct secondary structure propensity of the modification sites. Phosphorylated proteins are widely distributed in both disordered and folded proteins. Most of the proteins havd a folded conformation and only a small number of proteins were disordered (>50% disordered region) (shown in Figure 1A). In all three kinds of phosphorylated proteins, secondary structures were distributed similarly among the helix, sheet and coil. The helix and sheet take up around 40%, and the coil usually takes up the smallest part at 22~23% (Figure 1B). On the contrary, the secondary structure distribution of phosphorylated sites was significantly different. Most of the phosphorylation sites occurred in disordered regions; 52.18% of phosphotyrosine, 73.44% of phosphoserine, and 89.31% of phosphothreonine were located in disordered regions (Figure 1C). Considering the low percentage of coil structure in the overall structure, there is a clear tendency for phosphorylation to occur in disordered regions.

To sum up, phosphorylated proteins in PDB had a balanced distribution on major secondary structures, but phosphorylation sites tended to be located in disordered regions. Therefore, the precise sampling for disordered regions should be important. The findings suggested that a balanced force field for phosphorylated proteins’ simulation might achieve better accuracy in proteins with both folded and disordered regions.

### 3.2. Convergence of Simulation System

In order to evaluate the convergence, five independent trajectories of 100 ns each were obtained for each system. Convergence of the simulation needs to be considered first because it reflects the adequacy of the simulation time. We calculated the RMSE between the secondary chemical shifts of experimental value and the cumulative average value over time during the simulation. The results show that the simulation of the representative Ets1 system converges under *ff03CMAP*, *ff03*, *FB18*, and *CHARMM36m* force fields after 100 ns (shown in Figure 2). At the same time, we also compared the performance of sampling between multiple short trajectories (5*100 ns) and a single long trajectory (1000 ns). The results indicate that multiple short trajectories have better sampling ability than a single long trajectory (shown in Appendix A).

### 3.3. Phosphorylated Dipeptides

The simulations of phosphoserine, phosphothreonine and phosphotyrosine dipeptides behaved inconsistently in all four force fields. Using the FFscore as the metric, two force fields with *TIP4Pew* in phosphoserine, *ff03/TIP4Pew* in phosphothreonine, and *ff03CMAP/TIP4PD* in phosphotyrosine had the best performance. None of the four force fields performed optimally in all three types of phosphorylation (shown in Figure 3A).

In addition to chemical shifts, Ramachandran plots in Figure 3B presented the backbone dihedral angle distribution of all simulated dipeptides. TALOS-N [71]-predicted distributions from experimental chemical shifts were used as references. In reference Ramachandran plots, phosphoserine and phosphotyrosine were enriched in the PPII region, while phosphothreonine was enriched in the sheet region. Only simulation results from *ff03CMAP/TIP4PD* had a PPII- and sheet-like distribution, while those from other force fields had high helical propensity in phosphoserine and phosphothreonine.

In general, all force fields have similar accuracy and inconsistent performance in all three types of phosphorylated dipeptides, and *ff03CMAP* has a small advantage over *ff03* in force field scores and dihedral distributions. This is consistent with the previous studies of Vymetal J et al. [30], showing that force fields showed inconsistent performance on phosphorylated dipeptides. Therefore, further evaluation was performed for modified disordered proteins and folded proteins.

### 3.4. Phosphorylated Disordered Proteins

As discovered in statistical analysis through PDB, most phosphorylation sites were located in disordered regions. Further simulation tests on phosphorylated disordered proteins can serve as paradigms (references) for further (future) research. Two modified disordered proteins, tissue factor cytoplasmic tail (TF, PDB:2CEF) [53] and vitellogenin (Vg, PDB:2LID) [54], were selected as simulation targets.

Figure 4 shows the RMSD of chemical shifts, overall force field score (FFscore) and modification site force field score (Modification Score). Overall force field scores show that *ff03/TIP4PD* and *ff03CMAP/TIP4PD* were the best, while modification site force field scores preferred *ff03CMAP/TIP4PEw*. This elucidated that if we focus on the modification site, the performance of *ff03CMAP* is better than that of the *ff03* force field. Radius of gyration (Rg) and end-to-end distance distribution showed that four force fields have a huge difference in simulated Rg distribution (shown in Appendix A). *ff03/TIP4PEw* has the lowest average Rg which represents a more condensed status, while *ff03CMAP/TIP4PD* has the largest average and the widest range of Rg, which represents the extended structure of disordered proteins. In simulated ensembles’ secondary structure, *ff03CMAP* tends to have a more disordered structure than *ff03*, while the difference between *TIP4PEw* and *TIP4PD* is not significant (shown in Appendix A).

Sampling is another important factor for disordered protein simulation. The RMSD-Rg free energy landscape indicator can help to predict the sampling efficiency of different force fields. For disordered proteins, a wider energy landscape means the simulated structure will be more reasonable. As shown in Figure 3B, the energy landscape of *ff03/TIP4PEw* constrained in several conformations, while other force fields obtained a more adequate sampled conformation. With a better sampled landscape, *ff03CMAP* was able to obtain a more disordered conformation.

We compared the Ramachandran plot between simulation and the TALOS-N prediction from experimental data. The results show that *ff03CMAP/TIP4PD* has a propensity to be distributed in the PPII region, similar to the pattern found in phosphorylated dipeptides. This propensity has a superimposed effect, which means when we use the combination of *ff03CMAP* and *TIP4PD*, it will have a higher proportion in the PPII region. Analysis of hydrogen bonds also showed a difference between *ff03* and *ff03CMAP*, as *ff03CMAP* has less hydrogen bonds in all residues and modification sites (shown in Appendix A). After comparing chemical shifts, force field scores, Rg distributions, secondary structures, and energy landscapes, we can conclude that *ff03CMAP/TIP4PD* is the best system for phosphorylated disordered proteins.

### 3.5. Phosphorylated Folded Proteins and Complexes

Folded protein takes up the largest part of phosphorylated proteins. In nature, most protein function depends on specific structures, indicating that they are typically folded proteins with a defined, stable tertiary structure. In structural analysis, we found phosphorylation sites were usually located at the loop region (and also disordered regions) in a folded protein, or sometimes at a disordered protein which was bound to a folded protein. Therefore, we chose four folded proteins and two folded-disordered protein complexes as simulation targets for evaluation.

First, we compared the chemical shifts and force field scores. The overall force field scores of four force field systems had similar values and were close to 1. This means that from a holistic perspective, four force fields all have good performance, which is consistent with the previous works [35]. However, the modification site force field score was not unanimous among all four simulation systems. In folded modified proteins and folded-disordered protein complexes, *ff03CMAP* obviously had a better modification force field score than the *ff03* force field. In 2LKJ and 2LJE, *ff03CMAP/TIP4PEw* had the lowest modification site force field score, and in 2KMD and 5XK4, *ff03CMAP/TIP4PD* had the lowest modification site force field score.

In order to compare, two representative proteins were chosen for phosphorylated Ets1 (PDB: 2KMD) [57] and Ubiquitin (PDB: 5XK4) [58]. Ets1 is a reaction component of the Ras system. The two phosphorylated residues on Ets1 N terminal disordered region provide this motif with negative charge and repel its neighbor H0 motif from the negatively charged protein surface [57]. Therefore, the properties of the H0 helix should be key information for the simulation. In Figure 5A, we find that the secondary chemical shift for the H0 helix with *ff03CMAP/TIP4PD* has the best agreement with the experimental one among these force fields. RMSF (root mean square fluctuation) also shows that *ff03CMAP* has a relatively higher fluctuation than *ff03* simulated ensembles (shown in Appendix A). Additionally, the hydrogen bond analysis of Ets1 shows that *ff03CMAP/TIP4PD* had the least hydrogen bonds connected with two modification sites (shown in Appendix A). All of these results reflect that *ff03CMAP/TIP4PD* have a higher disorder propensity than other force fields, having secondary chemical shifts closer to zero, fewer hydrogen bonds and higher RMSF.

For phosphorylated ubiquitin, multiple sites of phosphorylation can act as a pH sensor in the cell [58]. These multi-phosphorylated ubiquitin can be found around mitochondria. In phosphorylated ubiquitin, its conformational switch between protonated and unprotonated form plays a key role in its function, and we chose unprotonated single phosphorylated ubiquitin as the target-simulated protein. The result of chemical shifts and force field scores indicated that phosphorylated ubiquitin had low force field scores and was similar to the folded protein Ets1. However, at the modified serine on the disordered region, the modification site force field score of *ff03CMAP/TIP4PD* was obviously better than other force fields (Figure 5B). In addition, concerning the Ramachandran plot of modification sites, we also discovered that *ff03/TIP4PEw*, which were the benchmark methods in previous research, formed an unusual distribution of dihedral angles (shown in Appendix A).

We also tested the performance of CHARMM36m [50] and FB18 [51] force fields on the folded protein Ets1 and the disordered protein TF, the chemical shift RMSD and force field scores of which are shown in Appendix A. The results show that FB18 had better performance than CHARMM36m for both systems, and close with the performance of ff03CMAP. For the disordered protein TF, ff03CMAP/TIP4PD had the lowest general force field score, while FB18 had the lowest modification force field score. For folded protein Ets-1, all force fields except for CHARMM36m had a relative optimal general force field score (less than 1.1), while ff03CMAP/TIP4PD had the lowest modification force field score.

## 4. Discussion

In our research, the main goal is to find an ideal force field for molecular dynamics simulations of post-translationally modified proteins. First, we studied the structural features of post-translationally modified proteins by analyzing the PDB protein structure database. We found that phosphorylation modifications, which occur most predominantly in post-translational modifications, have a distinct tendency to occur in the disordered regions of the protein. The conventional force fields for folded proteins sometimes might have inaccurate simulations of disordered proteins or disordered regions. Furthermore, previous studies have shown that existing force fields for post-translational modification are not accurate and consistent enough to simulate post-translationally modified peptides or proteins [29]. Therefore, we introduced the balanced *ff03CMAP* force field, which can perform well in molecular dynamics simulations of both folded and disordered proteins, to compare with the conventional ff03 force field.

By simulating a series of phosphorylated dipeptides, disordered proteins, folded proteins, and disordered-folded protein complexes, we summarized the results of the simulations of several force fields and focused on the discussion of the *ff03CMAP* force field and *ff03* force field with *TIP4PEw* and *TIP4PD* solvent models (shown in Figure 6).

The comparison with force fields, including *CHARMM36m* and *FB18* on folded proteins, indicates that the *ForceField_PTM* compatible *ff03* force field family is ideal for the simulation of phosphorylated proteins in a broader view of force fields. For the *ff03* force field family, the *ff03CMAP* force field showed improvement in simulation results of phosphorylated disordered proteins with less difference from experimental values compared with the *ff03* force field, and *TIP4PD* was better between the two solvent models. In folded proteins, the two force fields performed similarly, but when the post-translational modification site was located in the disordered region, the modification site simulation results were closer to the experimental values using the *ff03CMAP* force field. In the disordered-folded protein complexes, the test results were similar to those of the folded protein, and the post-translational modification site on the occurring disordered protein was closer to the experimental value in the simulation results of *ff03CMAP*. Overall, the simulations using the *ff03CMAP* force field outperformed the *ff03* force field when simulating biomolecules with post-translational modification sites located on disordered proteins or disordered regions.

Based on previous studies on potential charge in post-translational modification sites, it has been widely accepted that charge is an important component in the process of modeling post-translational modification sites [24,70]. In this research, our analysis reveals that the vast majority of phosphorylation occurs in the disordered region of the protein, which leads us to consider introducing CMAP energy for dihedral angles to improve the simulation of phosphorylated proteins. CMAP energy has been widely used to simulate disordered proteins [38,39,40,41,42,43,44] and can also be used to improve the balanced force field for disordered and structural proteins [35]. We have tried using the balanced force field to simulate phosphorylated proteins with both structured and disordered regions, and indeed the simulation of the modification sites in the disordered region was more accurate than other force field. This further illustrates that the methods used to develop disordered protein force fields can be used to improve the simulation of phosphorylated proteins.

However, the *ff03CMAP* force field also has limitations. Considering the simulation results of dipeptides, we found that neither the *ff03* force field nor the *ff03CMAP* force field was the consistent result for all three post-translational modifications. Similarly, in the protein simulations, we found that the dihedral angle distribution at the post-translational modification sites still notably differed from the predicted dihedral angle distribution. *ff03CMAP* still needs further improvement in the dihedral angle distribution at the post-translational modification sites.

## Figures and Tables

**Figure 1 ijms-23-11285-f001:**
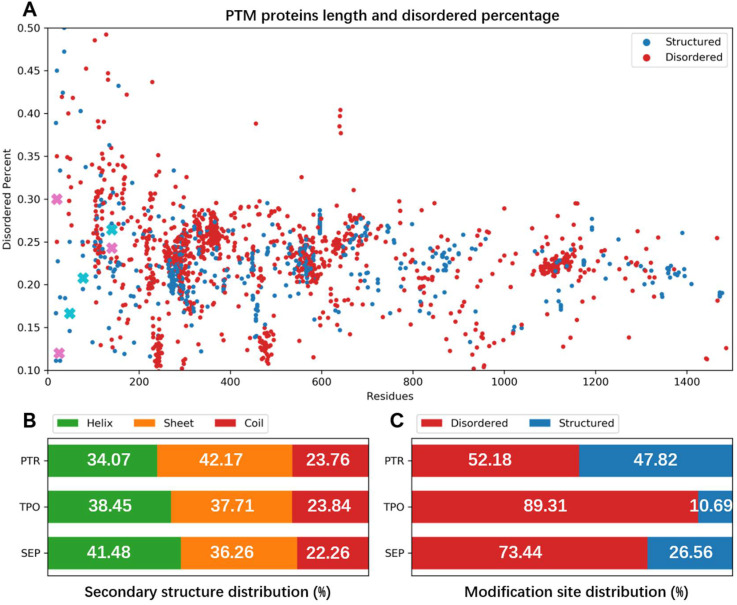
Structural analysis of phosphorylated proteins in PDB. Distributions of phosphorylated proteins’ disorder percentage and protein length (**A**), total protein secondary structure (**B**) and modification site secondary structure (**C**). In (**A**), each point represents a phosphorylated protein in the PDB database, and its color shows whether it had a phosphorylated residue in a disordered region (red) or not (blue). Selected simulation systems are marked by a cross in pink (modification in disordered region) and cyan (modification in folded region). In (**B**), the white numbers show the three kinds of secondary structures’ percentage in all residues in modified proteins. In (**C**), we present the two kinds (helix and sheet are considered as structured) of secondary structures’ percentage in all three kinds of phosphorylation site. (**B**,**C**) show that the phosphorylation is usually located in disordered regions.

**Figure 2 ijms-23-11285-f002:**
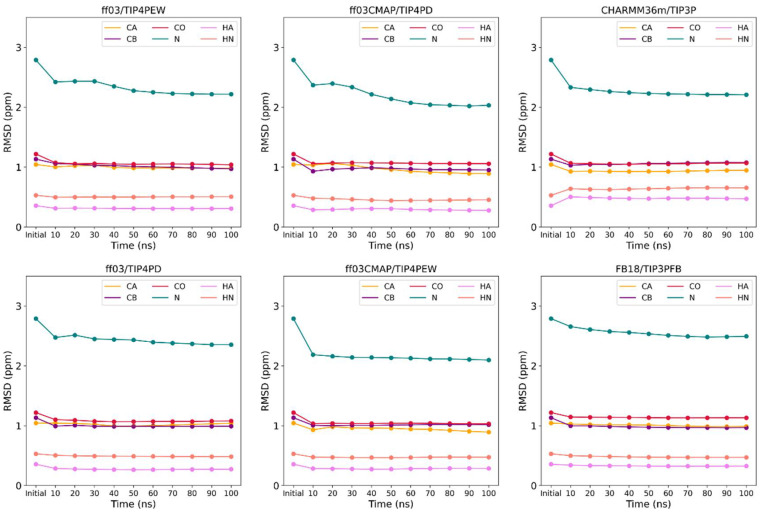
Time-dependent RMS error of chemical shift for Ets1 between experiment data and simulated ones with six force fields.

**Figure 3 ijms-23-11285-f003:**
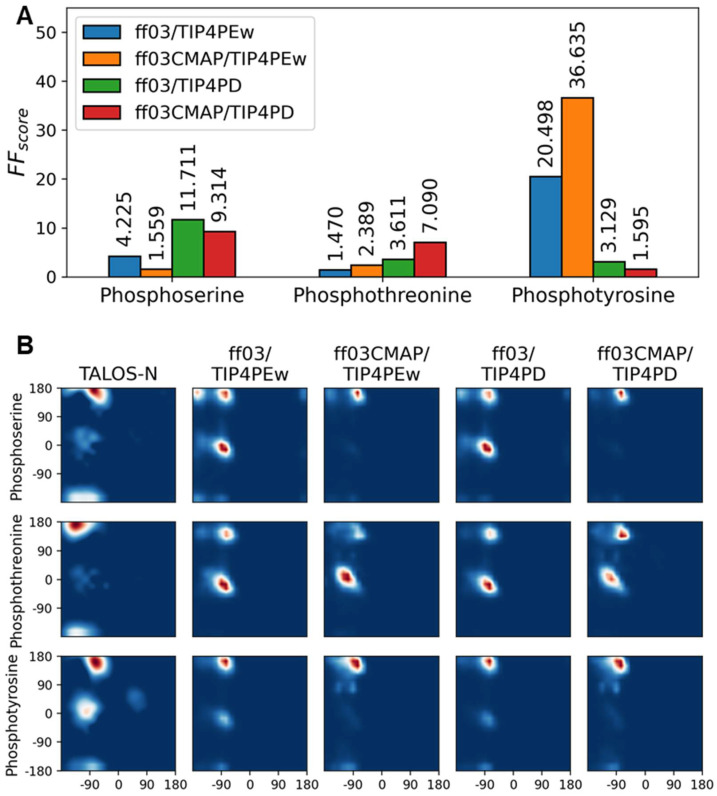
Simulation results of phosphorylated dipeptides. Comparison of force field scores (**A**) present the inconsistency of different force fields. Predicted and simulated Ramachandran plots (**B**) showed simulated distribution of backbone dihedral angle distribution difference between experiments and simulations.

**Figure 4 ijms-23-11285-f004:**
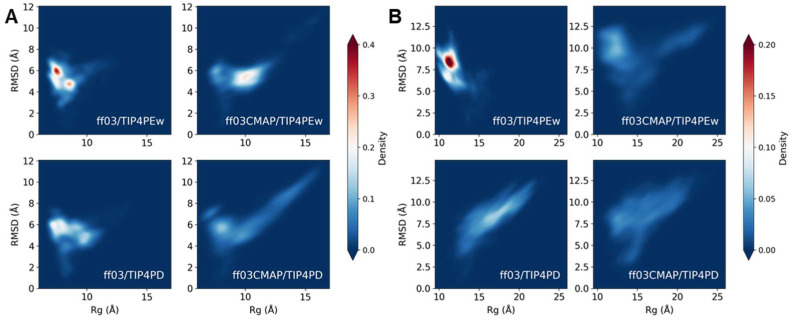
Energy landscape in MD simulation of phosphorylated disordered proteins. Comparison of different force field energy landscapes of phosphorylated disordered protein TF (**A**) and Vg (**B**) showed sampling efficiency advantage of *ff03CMAP* force field with both solvent models.

**Figure 5 ijms-23-11285-f005:**
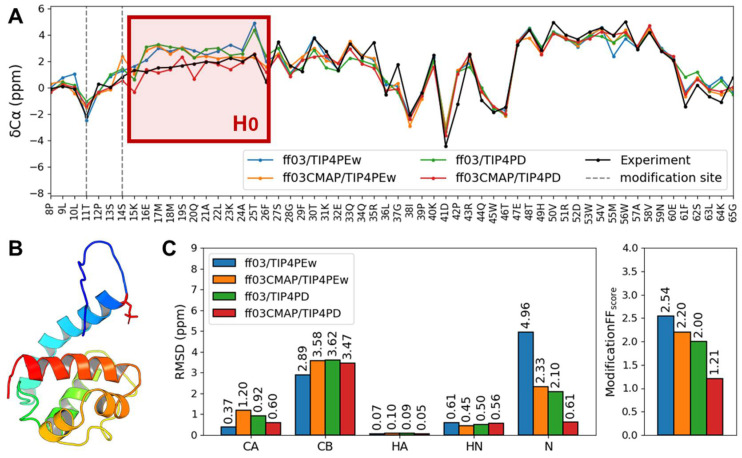
Simulation results of phosphorylated folded protein Ets1. Secondary chemical shifts of Ets1 around H0 region (**A**) showed only *ff03CMAP/TIP4PD* simulated the H0 helix best, whose conformation was greatly affected by close-site phosphorylations. Ets1 structure and its phosphorylation sites are shown in (**B**). Chemical shift RMSD of modified residues and modification force field scores (**C**) also support that *ff03CMAP/TIP4PD* performed best in phosphorylated residues.

**Figure 6 ijms-23-11285-f006:**
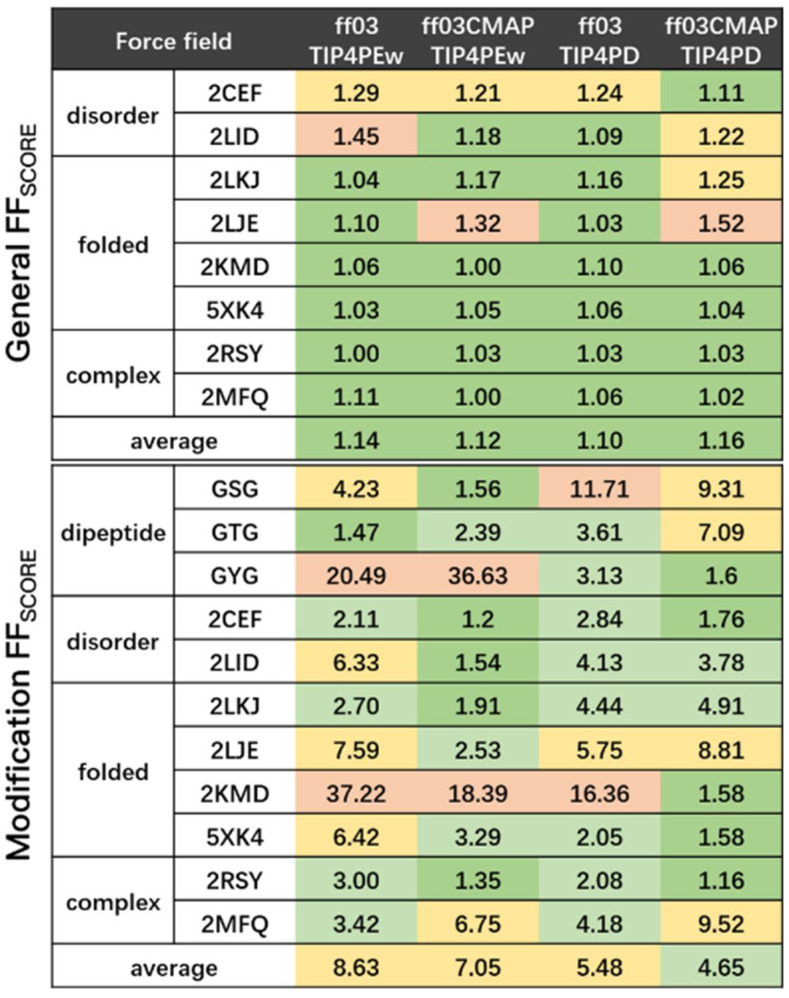
Overall force field score and modification site force field score of all simulated systems. Green represents a General FFscore lower than 1.2 and Modification score lower than 2. Light green represents a Modification FFscore between 2 and 5. Yellow represents a General FFscore from 1.2 to 1.3 and Modification FFscore from 5 to 10. Red represents a General FFscore higher than 1.3 and Modification Ffscore higher than 10.

**Table 1 ijms-23-11285-t001:** All simulated phosphorylated systems and simulation conditions.

System	Description	Modification Type	Length	Initial Structure	Ions	Number of Waters	Simulation Time (ns)	Number of Tested Force Fields	Number of Trajectories
**Dipeptide**									
GpSG [52]	Phosphoserine dipeptide	pSer	3	Extended	2 Na^+^	962	500	4	1
GpTG [52]	Phosphothreonine dipeptide	pThr	3	Extended	2 Na^+^	811	500	4	1
GpYG [52]	Phosphotyrosine dipeptide	pTyr	3	Extended	2 Na^+^	1089	500	4	1
**Disordered Protein**									
TF [53]	Cytoplasmic Tail of Tissue Factor	2*pSer	19	2CEF	2 Na^+^	2511	100	6	5
Vg [54]	Vitellogenin	pSer	35	2LID	7 Na^+^	6316	100	4	5
**Folded Protein**									
β3 [55]	β3 cytoplasmic tail	pSer	24	2LKJ	1 Na^+^	3303	100	4	5
αM [56]	αM cytoplasmic tail	2*pTyr	47	2LJE	2 Na^+^	3828	100	4	5
Ets1 [57]	Ets1	pSer, pThr	111	2KMD	9 Na^+^	7644	100	6	5
							1000	4	1
p-Ubiquitin [58]	Phosphorylated ubiquitin	pSer	76	5XK4	2 Na^+^	3613	100	4	5
**Complex**									
SH2 [59]	SH2 domain of Csk in complex with a phosphopeptide from Cbp	pTyr	137	2RSY	7 Na^+^	8267	100	4	5
TrkB [60]	FRS2a PTB domain with neurotrophin receptor TrkB	pTyr	138	2MFQ	7 Na^+^	8756	100	4	5

## Data Availability

Not applicable.

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
