# Peer review of "Balanced Force Field ff03CMAP Improving the Dynamics Conformation Sampling of Phosphorylation Site"

_ijms, 2022, doi:10.3390/ijms231911285_

Round 1

Reviewer 1 Report

July 28, 2022

Manuscript Number: ijms-1837145

Title: Balanced Force Field ff03CMAP Improving the Dynamics Con- 2 formation Sampling of Phosphorylation Site

Bozitao Zhong and Hai-Feng Chen

The authors evaluated the accuracy of the amber ff03 force filed and its variant (ff03CMAP they proposed) for phosphorylated proteins using two different water models (TIP4P-Ew or TIP4P-D). They suggest that ff03CMAP with TIP4PD is an option for simulating phosphorylated proteins, but neither ff03 nor ff03CMAP provides consistent performance across the tested systems. The results are useful for future simulations of phosphorylated proteins, as phosphorylation-focused assessments have never been done in the past to my knowledge.

The strength of their work is to present evaluation results focused on phosphorylation, but their comparison is limited to the ff03 and ff03CMAP parameter sets. Their conclusions to be supported by the presented data, but the data is not properly presented. In addition, the lack of detailed explanations in the figures and tables makes the understanding difficult. In summary, this manuscript is publishable, but we recommend the following minor revisions.

1) The experimental data to be used must be described in SI. In their previous publication (Zhang et al. J. Chem. Theory Comput. 2019, 15, 12, 6769–6780), the data used as metrics were listed in SI. The authors need to do the same here.

2) In this work, what is the number of classes of experimental measurements in equation (1)? 

3) Figure 1A needs to be explained in more detail. Is the following correct? (1) collected PDBs of phosphorylated proteins, (2) divided them into “structured” and “disordered”, (3) calculated the disordered %, and (4) sorted by protein length and plotted. How did you divide into “structured” and “disordered”? What are bold cross marks of different colors? Are (B) and (C) for entire PDBs (including both “structured” and “disordered”)?

4) Is Figure 2B a free-energy landscape or just a distribution? These figures should display tick marks, labels, and the bar showing the energy scale.

5) Is Figure 3 also a free-energy landscape?

6) In Figure 5, what does the background color mean?

7) What are the possible reasons for the better performance of ff03CMAP over ff03? A previous report by Robustelli (Ref. 55, Proc. Natl. Acad. Sci. 2018, 115, E4758–E4766) showed that changing the charge parameters improved the force filed accuracy. In the present case, the dihedral angle parameter correction (CMAP term) seems to be important, not non-bonded parameters. The authors can add a discussion on this point in comparison with previous works.

Author Response

Reviewer 1

The authors evaluated the accuracy of the amber ff03 force filed and its variant (ff03CMAP they proposed) for phosphorylated proteins using two different water models (TIP4P-Ew or TIP4P-D). They suggest that ff03CMAP with TIP4PD is an option for simulating phosphorylated proteins, but neither ff03 nor ff03CMAP provides consistent performance across the tested systems. The results are useful for future simulations of phosphorylated proteins, as phosphorylation-focused assessments have never been done in the past to my knowledge.

The strength of their work is to present evaluation results focused on phosphorylation, but their comparison is limited to the ff03 and ff03CMAP parameter sets. Their conclusions to be supported by the presented data, but the data is not properly presented. In addition, the lack of detailed explanations in the figures and tables makes the understanding difficult. In summary, this manuscript is publishable, but we recommend the following minor revisions.

1) The experimental data to be used must be described in SI. In their previous publication (Zhang et al. J. Chem. Theory Comput. 2019, 15, 12, 6769–6780), the data used as metrics were listed in SI. The authors need to do the same here.

Author reply: The experimental data to be used are now listed in supplementary Table S1.

2) In this work, what is the number of classes of experimental measurements in equation (1)?

Author reply: It depends on the protein we simulated with. We used all chemical shifts available through experiments. It should be a subset of C, Ca, Cb, Ha, HN, N. For example, some proteins only have HA and HN chemical shift, we will only use Ha and HN in equation (1).

3) Figure 1A needs to be explained in more detail. Is the following correct? (1) collected PDBs of phosphorylated proteins, (2) divided them into “structured” and “disordered”, (3) calculated the disordered %, and (4) sorted by protein length and plotted. How did you divide into “structured” and “disordered”? What are bold cross marks of different colors? Are (B) and (C) for entire PDBs (including both “structured” and “disordered”)?

Author reply: We have rewritten this part in our paper and updated the Figure 1.

4) Is Figure 2B a free-energy landscape or just a distribution? These figures should display tick marks, labels, and the bar showing the energy scale.

Author reply: Figure 2B (now is Figure 3B) is just a distribution. Besides, all free-energy landscape (Figure 4) figures are updated, we added the bar showing the density to present the energy scale.

5) Is Figure 3 also a free-energy landscape?

Author reply: Yes, this figure is also been updated.

6) In Figure 5, what does the background color mean?

Author reply: The background shows the magnitude of Force Field Score. Green represents low FFscore and Red for high FFscore.

7) What are the possible reasons for the better performance of ff03CMAP over ff03? A previous report by Robustelli (Ref. 55, Proc. Natl. Acad. Sci. 2018, 115, E4758–E4766) showed that changing the charge parameters improved the force filed accuracy. In the present case, the dihedral angle parameter correction (CMAP term) seems to be important, not non-bonded parameters. The authors can add a discussion on this point in comparison with previous works.

Author reply: We added a paragraph in discussion for this part. P19-20.

Reviewer 2 Report

Authors selected two variants of force fields coupled with two variants of solvents to test which provides better results for phosphorylated proteins. Although the work in important, as force fields for phosphorylated systems are far from perfect, the manuscript in the current state does not provide much information due to very limited set of force-field combinations and short simulations. Another force field benchmark published in the same journal a year ago uses 1000 ns simulations (instead of 100ns in this manuscript), proving that the longer time-scale is needed to capture complete dynamics of the systems (Int J Mol Sci. 2021 Sep; 22(18): 10174.). To make this manuscript publishable in IJMS, authors should add at least two more force fields (e.g. CHARMM36m and newly designed FB18 (J. Phys. Chem. B 2021, 125, 43, 11927–11942)) and make the trajectories significantly longer (due to the use of the GPU acceleration that authors mentioned in the text, for such small proteins, a microsecond timescale for a trajectory should be reached within couple of days). Moreover, language should be corrected, as there are numerous typos and grammar errors (e.g., in abstract “obvious better”) and details about performed simulations should be carefully rewritten to avoid confusion and errors:

  -  In the manuscript authors compare water models, but there is no information how many water molecules were used in the simulations and if and what size of periodic box was used.

  -  There are no tests if systems converged and if the PBC was big enough to prohibit interactions between mirror images.

   - 10ps simulations to obtain proper density may be too short (Amber authors advises nanosecond timescale for that) so authors should provide the densities that system reached after equilibration.

   - Authors do not explain why they selected these two water models for test, and not e.g., OPC water model, which is a default one in current version of Amber force field.

   - Time provided in Table 1 is misleading, as each trajectory consisted of only 100ns.

   - The distance for PME cutoff is not provided.

   - There is no need to provide temperature of the simulation in Table 1, as it is the same for all systems, however the exact ionic strength (and number and types of ions in each simulation box) is not provided, and sentence “The simulation temperature and ion strength were set according to their respective experimental conditions” is not correct, as e.g., 2LKJ was obtained at 308K, not 298K. Moreover, authors should decide if they want to mimic experimental or physiological conditions.

Author Response

Reviewer 2

Authors selected two variants of force fields coupled with two variants of solvents to test which provides better results for phosphorylated proteins. Although the work in important, as force fields for phosphorylated systems are far from perfect, the manuscript in the current state does not provide much information due to very limited set of force-field combinations and short simulations. Another force field benchmark published in the same journal a year ago uses 1000 ns simulations (instead of 100ns in this manuscript), proving that the longer time-scale is needed to capture complete dynamics of the systems (Int J Mol Sci. 2021 Sep; 22(18): 10174.). To make this manuscript publishable in IJMS, authors should add at least two more force fields (e.g. CHARMM36m and newly designed FB18 (J. Phys. Chem. B 2021, 125, 43, 11927–11942)) and make the trajectories significantly longer (due to the use of the GPU acceleration that authors mentioned in the text, for such small proteins, a microsecond timescale for a trajectory should be reached within couple of days). Moreover, language should be corrected, as there are numerous typos and grammar errors (e.g., in abstract “obvious better”) and details about performed simulations should be carefully rewritten to avoid confusion and errors:

Author reply: Indeed, the comparison of ff03 and ff03CMAP is made keeping the parameters for phosphorylation the same (FFPTM), while tests for CHARMM36m and newly developed FB18 are actually changing the parameters for phosphorylation, which is not very consistent with our main goal. But to further prove our conclusion, we added more force fields including C36m and FB18 in the evaluation (Figure S8).

In addition, we compared the sampling size between 5 parallel 100ns simulation and 1000ns simulation. The results show that a 5*100ns simulation can have similar or better sampling size than 1*1000ns simulation (Figure S1). Besides, we corrected the typo errors.

  -  In the manuscript authors compare water models, but there is no information how many water molecules were used in the simulations and if and what size of periodic box was used.

Author reply: We used octahedron boxes of solvent models with 10 Å to the edge (see the revised Materials and Methods: Molecular dynamics simulation).

  -  There are no tests if systems converged and if the PBC was big enough to prohibit interactions between mirror images.

Author reply: We added Figure 2 to illustrate that 100ns simulation is sufficient for the convergence. We added a water box 10 Angstrom larger than the protein size, which mean that the PBC should be big enough to prevent interaction between mirror images.

   - 10ps simulations to obtain proper density may be too short (Amber authors advises nanosecond timescale for that) so authors should provide the densities that system reached after equilibration.

Author reply: We have checked the density in our simulation, which means that the density of our simulated system is in relative stable in simulation. Also, we actually performed 100ps NPT simulation at the second heating step (see the revised Materials and Methods: Molecular dynamics simulation), where the density has already reached equilibrate.

   - Authors do not explain why they selected these two water models for test, and not e.g., OPC water model, which is a default one in current version of Amber force field.

Author reply: TIP4PEw and TIP4PD solvent model have been widely tested in simulating folded proteins and disordered proteins. For ff03 and ff03CMAP, these solvent models are the best ones to simulate proteins [Zhang, Y., et al., Well-Balanced Force Field ff03CMAP for Folded and Disordered Proteins. Journal of chemical theory and computation, 2019. 15(12): p. 6769-6780].

   - Time provided in Table 1 is misleading, as each trajectory consisted of only 100ns.

Author reply: We updated the Table 1.

   - The distance for PME cutoff is not provided.

Author reply: It’s now provided in main text.

   - There is no need to provide temperature of the simulation in Table 1, as it is the same for all systems, however the exact ionic strength (and number and types of ions in each simulation box) is not provided, and sentence “The simulation temperature and ion strength were set according to their respective experimental conditions” is not correct, as e.g., 2LKJ was obtained at 308K, not 298K. Moreover, authors should decide if they want to mimic experimental or physiological conditions.

Author reply: We simulated all our systems at room temperature, corresponding information is corrected in main text.

Round 2

Reviewer 2 Report

Authors significantly improved the manuscript, however, some issues remained unsolved and should be corrected or explained:

- Authors should prove that PBC box was big enough to prohibit interactions between periodic images by presenting minimum distance between periodic images vs time (there is a special tool for that included in ccptraj) - solvation sphere of 10A may be not enough if the disordered part of the protein stretch.

 - Authors should expand the abstract by adding one more sentence concluding results obtained with CHARMM36M and FB18 force fields.

- Authors did not discuss how adding physiological concentration of ions (e.g. 0.15M of NaCl) would impact the results. It was postulated many times that ionic concentration can have significant impact especially on disordered proteins.

- As Figure S9 shows that FB18 is more accurate than other tested force fields for all residues of 2CEF system, authors should at least briefly discuss it in the manuscript.

- Authors claim "Considering that all popular major force fields (ff99SBildn32, ff0333 and 106 CHARMM36m34) are more suitable for folded proteins" is somehow controversial, as the title of the CHARMM36m paper is "CHARMM36m: an improved force field for folded and intrinsically disordered proteins".

- Statement "Compared with ff03 force field, ff03CMAP adds a correction map (CMAP) parameter" is correct, but for clarity authors should mention that all modern versions of the all-atom force fields contain CMAPs (e.g. Amber ff19sb) and their purpose is not only to improve disordered regions of proteins, as authors suggest in Introduction.

- Statement "The next steps were performed in the pmemd, and the 5 dependent trajectories of each 100ns MD simulations were performed with pmemd.cuda" is inconsistent with the Table 1.

- Figures S8 and S9 legend should be "CHARMM36m" not just "CHARMM" or "CHARMM36" and all force field names with combination with water models should be identical in all labels, legends and text in both main manuscript and SI.

- There are still typos or other text errors which should be corrected (e.g. Table 1 "Simulation Time/ns" - simulation time is not divided by ns, so it should be "Simulation Time [ns]"; lack of plural form in "Number of Tested Force Field").

Author Response

Prof. Hai-Feng Chen

Tel: 86-21-34204073

Fax: 86-21-34204073

Shanghai, Sept. 08, 2022

Manuscript Number  ijms-1837145R1

Dear Mr. Carlie Chen,

Please find the revised manuscript entitled “Balanced Force Field ff03CMAP Improving the Dynamics Conformation Sampling of Phosphorylation Site”. We thank you and the reviewer for the time and effort in reviewing the previous version of the manuscript. In the following, we list our revision efforts addressing the reviewers’ comments and refining the language. The revised portions of the manuscript are also marked in red to highlight the changes.

Reviewer  

Authors significantly improved the manuscript, however, some issues remained unsolved and should be corrected or explained:

- Authors should prove that PBC box was big enough to prohibit interactions between periodic images by presenting minimum distance between periodic images vs time (there is a special tool for that included in ccptraj) - solvation sphere of 10A may be not enough if the disordered part of the protein stretch.

Author reply: Minimum distance between periodic images vs time of a disordered protein is now presented in supplementary Figure S1. We also clarify this in text. P8.

 - Authors should expand the abstract by adding one more sentence concluding results obtained with CHARMM36M and FB18 force fields.

Author reply: We added one sentence to state the results of CHARMM36m and FB18 in abstract.

- Authors did not discuss how adding physiological concentration of ions (e.g. 0.15M of NaCl) would impact the results. It was postulated many times that ionic concentration can have significant impact especially on disordered proteins.

Author reply:The effect of ions concentration is stated in manuscript with citations (Molecular Dynamics Simulation). P8.

- As Figure S9 shows that FB18 is more accurate than other tested force fields for all residues of 2CEF system, authors should at least briefly discuss it in the manuscript.

Author reply: We added discussion of FB18 and CHARMM36m result for 2 additional tested systems.

- Authors claim "Considering that all popular major force fields (ff99SBildn32, ff0333 and 106 CHARMM36m34) are more suitable for folded proteins" is somehow controversial, as the title of the CHARMM36m paper is "CHARMM36m: an improved force field for folded and intrinsically disordered proteins".

Author reply: We modified this sentence by replacing CHARMM36m with ff14SB.

- Statement "Compared with ff03 force field, ff03CMAP adds a correction map (CMAP) parameter" is correct, but for clarity authors should mention that all modern versions of the all-atom force fields contain CMAPs (e.g. Amber ff19sb) and their purpose is not only to improve disordered regions of proteins, as authors suggest in Introduction.

Author reply: We mentioned the pervasive using of CMAP in the new manuscript.

- Statement "The next steps were performed in the pmemd, and the 5 dependent trajectories of each 100ns MD simulations were performed with pmemd.cuda" is inconsistent with the Table 1.

Author reply: We modified this sentence, now it’s more accurate to describe the simulation times.

- Figures S8 and S9 legend should be "CHARMM36m" not just "CHARMM" or "CHARMM36" and all force field names with combination with water models should be identical in all labels, legends and text in both main manuscript and SI.

Author reply: We modified the figures and corresponding sentences. Force field names are now identical anywhere presented, and all in italic font style in the text.

- There are still typos or other text errors which should be corrected (e.g. Table 1 "Simulation Time/ns" - simulation time is not divided by ns, so it should be "Simulation Time [ns]"; lack of plural form in "Number of Tested Force Field").

Author reply: We corrected our typos in main manuscript and SI.

Haifeng Chen, Ph.D.

Professor of Bioinformatics and Biostatistics

Round 3

Reviewer 2 Report

Authors answered all of the questions and made all corrections except for correcting CHARMM36m force field name in legend of Figures S9 and S10. In the present form, manuscript should be a valuable contribution to the scientific community.